# Hydrothermal Reduction of CO_2_ to Value-Added Products by In Situ Generated Metal Hydrides

**DOI:** 10.3390/ma16072902

**Published:** 2023-04-06

**Authors:** Xu Zeng, Guodong Yin, Jianfu Zhao

**Affiliations:** State Key Laboratory of Pollution Control and Resources Reuse, College of Environmental Science and Engineering, Tongji University, 1239 Siping Road, Shanghai 200092, China; yinguodong@tongji.edu.cn (G.Y.); zhaojianfu@tongji.edu.cn (J.Z.)

**Keywords:** hydrothermal reduction, CO_2_, hydrogen storage, metal hydrides, hydrogenation

## Abstract

An integrated process by coupling hydrothermal reactions, including CO_2_ reduction and H_2_O dissociation with metals, is proposed. The hydrogen could be rapidly produced under hydrothermal conditions, owing to the special characteristics of high temperature water, generating metal hydrides as intermediates. Hydrogen production from the H_2_O dissociation under hydrothermal conditions is one of the most ideal processes due to its environmentally friendly impact. Recent experimental and theoretical studies on the hydrothermal reduction of CO_2_ to value-added products by in situ generated metal hydrides are introduced, including the production of formic acid, methanol, methane, and long-chain hydrocarbons. These results indicate that this process holds promise in respect to the conversion of CO_2_ to useful chemicals and fuels, and for hydrogen storage, which could help alleviate the problems of climate change and energy shortage.

## 1. Introduction

The utilization of renewable resources is considered as one of the most promising pathways due to humankinds’ vast demands for energy and resources [1,2,3]. However, large amounts of energy are necessary in order to utilize renewable resources [4,5]. Hydrothermal hydrogen generation from water is one of the most ideal methods due to its environmental friendly impact [6,7]. Nevertheless, the utilization of hydrogen has high storage and transportation costs because hydrogen is gaseous under ambient conditions. Extensive studies of potential hydrogen storage solutions have been examined, such as metal hydrides and organic frameworks [8]. However, the efficiency is always low, and the systems are complicated. High efficiency remains a big challenge [9].

The popular term “hydrothermal” has been used broadly to refer to water under high temperature and high pressure. Hydrothermal reactions have been considered as an important process in the formation of fossil fuels [10,11]. Generally, the hydrothermal alteration of minerals could induce the formation of abiotic chemicals, which arise from the dissolved CO_2_. The abiotic synthesis of organics means that CO_2_ reduction could be realized with metals under hydrothermal conditions. Then, an integrated process by coupling hydrothermal reactions, including CO_2_ reduction and H_2_O dissociation, with metals is proposed, as shown in Figure 1. In the right half of the circular process, metal oxides could be reduced by the renewable thermochemical reduction process. As we know, the rapid increase of the concentration of CO_2_ in the air has caused some adverse effects for humankind and the environment [12,13,14]. Thus, if a cycle that included hydrogen storage and CO_2_ conversion to organics [15] was achieved industrially, the concept of hydrogen storage and CO_2_ utilization could help alleviate the problems of climate change and energy shortage.

## 2. Characteristics of Water under Hydrothermal Conditions

Hydrothermal reactions are important because water possesses some special characteristics under hydrothermal conditions. In the following sections, the representative characteristics of water under high-temperature and high-pressure conditions are introduced.

### 2.1. Ion Product

The product of the H^+^ and OH^−^ concentrations is denoted as the ion product (*K_w_*), the unit of which is mol^2^/kg^2^. As the temperature increases, the ion product increases from *K_w_* = 10^−14^ mol^2^/kg^2^ at about 20 °C to approximately 10^−11^ mol^2^/kg^2^ at around 300 °C [16]. When the *K_w_* value is bigger, water may show enhanced acidic or basic catalytic activity for reactions, owing to the high H^+^ and OH^−^ concentrations [17]. Therefore, the hydrogen production under hydrothermal conditions would be influenced by the H^+^ ion, especially under high temperature.

### 2.2. Water Density

Water density is another important property that can change significantly with temperature and pressure. When the temperature increases, the water density decreases. Water density decreases from about 800 kg/m^3^ to about 150 kg/m^3^ during the transition from liquid phase to gas phase. A change of water density means a variation at the molecular level, including hydrogen bonding, dielectric strength, etc. [18] Therefore, we can see that the change of water density is also related to the temperature.

### 2.3. Dielectric Constant

The dielectric constant also decreases sharply when the water temperature increases. The dielectric constant is 78.5 under ambient conditions [19]. Water near the critical point has properties like organic solvents, which can dissolve some organic compounds. The advantage of supercritical water is that higher concentrations of reactants can be attained.

### 2.4. Hydrogen Bonding

The number of hydrogen bonds at different temperatures and densities is shown in Figure 1. It can be concluded that hydrogen bonding becomes weaker and less persistent while increasing the temperature and decreasing the density [20,21]. The hydrogen bonding network exists as large, percolating clusters under ambient conditions [22,23,24]. When the temperature increased and the density decreased, the average cluster size always decreased. These results indicated that less hydrogen bonding results in much less order than ambient conditions. Individual molecules can join the elementary reaction as a hydrogen source in the hydrothermal reactions.

## 3. Hydrogen Production Using Metal

Yavor et al. [25] introduced some hydrothermal reaction systems with metal powder. It was found that hydrogen production was related to the reaction temperature, including the properties of the metal powders. The authors reported that the metals, such as Al, Mn, and Mg, could serve as energy carriers for hydrogen storage. The theoretical amount of hydrogen could be calculated as shown in Equations (1) and (2):(1)$$xM+yH_{2}O\to \;M_{x}O_{y}+y\;H_{2}$$
(2)$$xM+2yH_{2}O\to \;xM{\left(OH\right)}_{2y/x}+yH_{2}$$

As shown in Figure 2, the theoretical hydrogen production for each metal is presented. The estimated activation energy is also calculated. The results reasonably fit the Arrhenius equation, with an approximately linear behavior. The slope of the trend line curve is similar to that of *E_a_*. These results suggested that hydrogen could be easily produced with metals under hydrothermal conditions, which can join the integrated process of the H_2_O dissociation and the CO_2_ reduction with the reduction of metal oxides to metals, as shown in Figure 1.

## 4. Experimental Studies of CO_2_ Reduction to Formic Acid

In this section, recent experimental studies on the hydrothermal reduction of CO_2_ to formic acid by Zn, Al, Fe, and Mn were introduced. These selected metals, which are not noble, are common and cheap. In addition, these metals exhibited high efficiency for hydrogen production, which can be seen in Section 3. Therefore, the selected metals show promise for the reduction of CO_2_. The detection of metal hydrides by IR analysis was also introduced in this section.

For the hydrothermal reduction of CO_2_, experiments were conducted using a series of batch SUS 316 tubing reactors with end fittings, providing an inner volume of 5.7 mL. NaHCO_3_ was used as a CO_2_ resource to simplify handling. The zero-valent metals were in powder form with a particle size of 200 mesh without any further treatment. All the reagents were purchased from Sinopharm Chemical Reagent Co., Ltd. (Shanghai, China), and were of analytical grade. The experimental procedure was conducted as follows [14]. The desired amounts of metal powder, NaHCO_3_, and deionized water were added to the reactor. Then, the reactor was sealed and put into a salt bath that had been preheated to desired temperature. After the preset reaction time, the reactor was taken out and placed into a cold water bath to quench the reaction. After cooling to room temperature, the reaction mixture was collected and filtered through a 0.22 μm syringe for analysis. The water filling was defined as the ratio of the volume of the water put into the reactor to the inner volume of the reactor, and the reaction time was defined as the duration of time that the reactor was kept in the bath.

### 4.1. CO_2_ Reduction by Zinc

Highly efficient hydrothermal reduction of CO_2_ to formate by zinc was reported [14]. As shown in Figure 3a, the highest yield of formate, approximately 80%, was acquired after 60 min at 325 °C. The yield reached up to about 45% after 10 min at 300 °C. At 250 °C, the yield, near 25%, was obtained after 10 min. These results illustrated that CO_2_ could be reduced to formate with high efficiency under hydrothermal conditions with zinc. The effects of reaction temperature illustrated that higher reaction temperature is very favorable for the production of formate. To verify that a reaction temperature of 325 °C was necessary for the high yield, experiments were conducted with a longer reaction time at 250 °C. As shown in Figure 3b, the formate yield was only about 60% with a reaction time of 400 min. These results suggest that the highest formate yield from each reaction was determined by the reaction temperature. Interestingly, these phenomena are related to the maximal ion-product constant of high-temperature water occurring between 280 and 300 °C [25,26], which suggests that the equilibrium was related to the characteristics of high-temperature water.

The oxidation product of zinc was subsequently investigated. As shown in Figure 4a, zinc was oxidized to ZnO after the hydrothermal reaction. In Figure 4b, it can be seen that all zinc was oxidized to ZnO after 10 min, which illustrated that zinc could be oxidized rapidly. It should be noted that almost all of the Zn was oxidized to ZnO within 10 min. After 10 min, the increased rate of formic acid yield decreased significantly. When the reaction of Zn with H_2_O was almost finished, the yield of formic acid increased slowly. Thus, it can be concluded that the efficient reduction of CO_2_ should be primarily attributed to the reaction of Zn with H_2_O.

### 4.2. Detection of Zinc Hydrides

The infrared spectrum method was adopted to detect the intermediates, to investigate the reaction mechanism in detail [14]. As shown in Figure 5a, a peak at 3336.8 cm^−1^ appeared by the stretching mode of the O–H complexes [27,28,29], which was clearly observed in the produced samples. However, almost no extra IR absorption peaks were found (see Figure 5b). These results suggest that the intermediate involves the structure of species of H–Zn…O–H.

Zinc hydrides have been reported to be a possible sources of active hydrogen for hydrogenation [30] and the synthesis of the mononuclear alkyl Zn-H complex [31]. Therefore, the IR absorption peak indicates that the zinc hydrides were produced as the intermediates, which gives some information for the explanation of the reaction mechanism.

### 4.3. CO_2_ Reduction by Aluminum

Al, which possesses an excellent ability to reduce the hydrogen production from H_2_O, is one of the most abundant metals in the crust of earth. Therefore, if CO_2_ could be reduced efficiently by the hydrogen produced with Al under hydrothermal conditions, a novel facile pathway for CO_2_ utilization could be developed, along with the utilization of abundant aluminum. Furthermore, the Al oxides also can be reduced to Al, which is similar to the reduction of ZnO by using concentrated solar energy. Therefore, Al exhibited promising prospects for the hydrothermal reduction of CO_2_.

Yao et al. studied the CO_2_ reduction by Al under hydrothermal conditions [32]. It was reported that Al can enhance the reduction of NaHCO_3_ to formate by in situ hydrogen. The highest yield of formate 64% was achieved. As shown in Figure 6, the formate yield significantly increased with the increase of reductant amount. However, the yield decreased gradually with the increase of NaHCO_3_ amount. The reason may be that the increase of the amount of NaHCO_3_ induced the decrease of the ratio of reductant and CO_2_. Therefore, the reductant amount should be sufficient to acquire a higher formate yield. These results illustrated that a suitable balance existed for high efficiency between the amounts of reductant and CO_2_.

The yield of formic acid increases with the increase of the reaction time (see Figure 7). Although the formic acid yield increased with longer time, 2 h is a better choice for considering the energy cost. As shown in Figure 7, the results also showed that formic acid yields increased significantly when the reaction temperature increased from 250 to 300 °C. It was very interesting that the yield decreased to 34% at 325 °C. The reason is probably that the decomposition of formic acid was sped up under higher temperature. Therefore, the preferred reaction temperature is 300 °C. The variation trend of the formic acid yields with Al is similar to the results of Zn. The reason for that may be the decomposition of the formate under higher temperature with Al and Zn. Therefore, it is better to control the reaction temperature at 300 °C. For the reaction time, perhaps the reaction time of 2 h is a better choice. In our previous study on the hydrothermal oxidation of biomass, we found that the suitable reaction temperature was also 300 °C. This was likely related to the special characteristics of water under high temperature.

### 4.4. CO_2_ Reduction by Iron

The metal Fe is one of the most abundant metals in the Earth’s crust. In our previous study, it was reported that iron oxide after oxidation can be reduced easily by bio-derived chemicals such as glycerin [26]. Therefore, the cycle could be realized when Fe was used for the hydrothermal reduction of CO_2_. For this reason, the experiments with Fe as a reductant for the CO_2_ reduction were performed [33]. As shown in Figure 8, kinetic curves of the formic acid production based on the parameters of reaction temperature and time were exhibited [34]. In the beginning, the yield increased slowly, and then increased rapidly with the extension of reaction time. At 300 °C and 325 °C, a high formic acid yield of 40~60% was achieved in the first 30 min, indicating that a higher temperature is favorable for the production of formic acid. Zhang et al. reported Fe-hydride formation in the selective Fe-catalyzed CO_2_ reduction in aqueous solution [35]. At present, Fe hydride is normally reported in the organometallic reaction for CO_2_ reduction. These studies give us some information about the reductant activity of Fe hydride.

### 4.5. CO_2_ Reduction by Manganese

Mn, a first-row transition metal, has special coordination chemistry due to its reactive redox nature. Mn can mediate the H_2_O dissociation to provide the necessary electrons for photosynthesis [36,37]. Besides, Mn plays an important role in the CO_2_ hydrogenation and Fischer−Tropsch synthesis [38]. Therefore, hydrothermal reduction of CO_2_ in the presence of Mn could be an effective method.

Lyu et al. reported highly selective reduction of CO_2_ into formic acid with Mn powder as a reductant [39]. The highest yield of formic acid 75% was acquired in the presence of Mn under hydrothermal conditions, as shown in Figure 9. In this process, CO_2_ not only acts as a carbon source, but also accelerates the hydrogen production from H_2_O. By coupling this process with the MnO/Mn cycle using solar energy, the CO_2_ recycle utilization would be realized. The effects of reaction temperature and time on the yield of formic acid were investigated. The yield of formic acid was initially very low, then increased rapidly with the reaction time increasing at 250 and 275 °C. The yield was 60% when the temperature reached 325 °C. Therefore, it could be concluded that higher temperature benefits the production of formic acid. These results illustrated that Mn addition is an effective method for highly efficient H_2_O dissociation for CO_2_ reduction.

Presently, the photocatalytic and electrocatalytic reduction of CO_2_ has been studied extensively. In comparison, hydrothermal reduction of CO_2_ coupled with the production of hydrogen through H_2_O dissociation by metal, along with the renewable thermochemical reduction of metal oxides, is a novel concept. Based on the introduction above, experimental results of CO_2_ reduction to formic acid by Zn, Al, Fe, and Mn showed that metals, which can produce hydrogen efficiently under hydrothermal conditions, show promise for the hydrothermal reduction of CO_2_. However, the detailed reaction mechanism is still unclear. The reaction mechanism is commonly studied with the hypothesis derived from the conformation of reactants and products. Because these reactions take place under high temperature and high pressure, the detection of reaction intermediates is not easy. Therefore, theoretical study could be used for the discussion of the mechanism.

## 5. Theoretical Study of CO_2_ Reduction to Formic Acid

Density functional theory (DFT) has always been utilized in the investigation of mechanism study of various types of chemical reactions [40,41]. In this section, recent theoretical studies on the hydrothermal reduction of CO_2_ by in situ produced hydrogen over metals were introduced.

### 5.1. CO_2_ Reduction by Zinc Hydrides

#### 5.1.1. Formation of Zinc Hydrides

In our previous study, we explored the potentials of metal hydrides as intermediates in the reaction with Zn and H_2_O [42]. A Zn_5_ cluster of the trigonal dipyramidal structure was constructed, see Figure 10. The vertex of Zn and their bonds are shown in black and gray. For the DFT calculation, five sites are adopted to generate initial geometries. Positions 1 to 5 were presented in detail in the literature. The decomposed five fragments of two H_2_O molecules are distributed. The decomposed five fragments from two waters are distributed as shown in Table 1. All the possible structures that two H_2_O molecules can have were generated with the symmetry of the Zn_5_ cluster, see Table 2. A total of 859 structures were generated. All the structures were optimized, using the PW91 functional, which considered the van der Waals interaction.

In the calculation of Zn_5_/fragments complexes, the most stable one is shown in Figure 11. Pattern (6) shows the most stable one with the lowest energy. Based on these calculation results, it was found that, in the hydrogen production process, the H_2_O molecule is decomposed to O^2−^ and two H^+^_._ The Zn-fragment stabilizes in the order of Zn–H_2_O, Zn–OH, Zn–O and Zn–H, forming Zn–H and Zn–O–Zn. When O^2−^ connects to Zn atoms, more than two bonds around O^2−^ were observed in the optimized structure. Therefore, the energy is relatively low, which indicated that O^2−^ atom binding to Zn is more stable than OH^−^ binding. Thus, we can conclude that zinc hydrides, not Zn–OH, are intermediate states. This conclusion could be used to explain different reactions, in particular, the reactions between metal and H_2_O.

It should be noted that it is possible to cover all the possible structures in the considered patterns, therefore the true minimum can be found with high probability. In addition, additional calculations on Zn_20_ cluster with ADMP sampling were carried out. This procedure could not guarantee the global minimum. However, the structure with the lowest energy still suggested a similar binding mode and energy gaps. Based on these considerations and calculations, it was concluded that the Zn_5_ possible intermediate in the hydrogen production reaction with Zn and H_2_O consists of the zinc hydrides species, which should be considered in the discussion of the reaction mechanism.

#### 5.1.2. CO_2_ Reduction by Zinc Hydrides

In our previous study, we reported a theoretical study to investigate the reaction mechanism [43]. From a thermodynamic point of view, two processes were compared by using available thermodynamic data [44].
(3)$$\mathrm{Zn}\;+\;H_{2}O\;+\;\mathrm{NaHC}O_{3}\to \;\mathrm{ZnO}\;+\;\mathrm{HCOONa}\;+\;H_{2}O\;\Delta G\;(573\;K)=-103.79\;\mathrm{kJ}\cdot {\mathrm{mol}}^{-1}$$
(4)$$\mathrm{ZnO}\;+\;H_{2}+\;\mathrm{NaHC}O_{3}\to \;\mathrm{ZnO}\;+\;\mathrm{HCOONa}\;+\;H_{2}O\;\Delta G\;(573\;K)=-5.37\;\mathrm{kJ}\cdot {\mathrm{mol}}^{-1}$$

For Equations (3) and (4), the Δ*G* value is negative, which means that high yield of formic acid was possible from the thermodynamics point of view. H–Zn…O–H species could be produced due to the chemisorbed H_2_ on ZnO [45]. Zinc hydride has been reported as an active reducing agent which can reduce CO_2_ [46]. For the reaction of Equation (3), Zn–H also was reported, which can be produced in considerable amounts in the reaction of Zn with H_2_O [47]. Then, the whole process is understandable. As shown in Figure 2, we proposed Zn–H species as a plausible intermediate, which played a crucial role in the reduction of HCO_3_^−^ for the production of formic acid. During the first step, Zn–H species was formed by the oxidation of Zn with water. Then, Zn–H complex, as an active reducing agent, reduces HCO_3_^−^ into HCOO^−^.

As shown in Figure 12, the activation energy to the transition state from the initial state of Zn–H and HCO_3_^−^ is 24.1 kcal·mol^−1^. The geometry and the Mulliken charge at the transition state in the production of HCOO^−^ from HCO_3_^−^ are also shown in Figure 12. The Zn–H bond distance is about 1.94 Å. The charge of H in the Zn–H specie is −0.221. Note that this value is that of a hydride rather than a proton. When the H of Zn–H attached to the C of HCO_3_^−^, the C–H bond would be produced by through the vibration of C. Afterwards, various ways for HCO_3_^−^ to approach the Zn–H area were examined to locate the TS shown in Figure 12. In the end, the OH^−^ escaped from the Zn_5_ cluster. These results were reported for the first time in the hydrothermal reduction of CO_2_ with Zn as a reductant. The Zn_5_ cluster is not big, however, as mentioned above, all the possible structures in the considered patterns of Zn + H_2_O were considered, therefore the true minimum can be found with high probability.

As shown in Figure 13, the IRC calculation and shapes of the HOMO and LUMO of TS are presented. IRC from the TS leads smoothly to the reactant Zn–H + HCO_3_^−^, as well as to the formation of formic acid. The occupied HOMO of TS shows a bonding interaction between the H of the zinc hydride and the C of HCO_3_^−^, whereas the unoccupied LUMO of TS shows an antibonding interaction between Zn and the H of zinc hydride. This means that at the TS, the Zn–H bond will be cleaved, as an unoccupied orbital and simultaneous C–H bond was formed. These results verified that this could be a possible reaction mechanism for the production of formate based on the intermediate of zinc hydrides, which were produced in the hydrogen gas production. Based on these discussions, it can be suggested that zinc hydride (Zn–H) is a key intermediate species in the reduction of CO_2_ to formic acid, which demonstrates that the formation of formic acid is through an SN2-like mechanism.

### 5.2. CO_2_ Reduction by Aluminum Hydrides

A theoretical study was conducted to investigate the hydrothermal reduction of CO_2_ to formic acid using Al by DFT calculations. B3LYP functional using the extended 6–311+G (3df, 2p) basis set was adopted for the calculation. The formation of H-Al-OH species, an intermediate in the reaction of Al with H_2_O, was calculated firstly. The formation of HCOO^-^ was simulated subsequently. Results showed that Al-H is a key intermediate species, which demonstrated that the formation of formic acid is through an SN2-like mechanism, as well [48]. The geometries and Mulliken charge could be seen in Figure 14. As shown in Figure 14, in the start state, when HCO_3_ adsorbed onto an Al atom with two oxygen atoms, the Mulliken charge of the H atom in Al-H was negative, which means that the H was a proton. However, in the transition state and final state, the Mulliken charge of the H atom became positive. These results illustrated that metal hydrides have strong reactant activities. The calculated energy results showed that the final state is more stable (with respect to initial state) by −7.31 kcal/mol.

In Figure 15, the IRC calculation and HOMO, LUMO orbital of TS are presented. As shown in Figure 15, the IRC from TS smoothly leads to the reactant of Al-H+HCO_3_^−^, which means the formation of formic acid. These results were similar to those of Zn. The unoccupied HOMO and LUMO of TS suggested an antibonding interaction between Al and H atoms of the Al-H species. The Al-H bond will be cleavage, unoccupied orbital, and simultaneously, the new C-H bond will be formed. The generation of hydrogen gas was not studied in this literature; even the calculation of HCOO^-^ formation was still a little rough due to the small calculation system.

### 5.3. CO_2_ Reduction by Iron Hydrides

In our previous study on CO_2_ reduction by Fe, it was reported that H_2_O is used as a source of hydrogen and Fe is used as a reductant, which can rapidly produce hydrogen. With the in situ produced hydrogen, CO_2_ was reduced to formic acid efficiently. After the reaction, Fe was oxidized to Fe_3_O_4_. It was also reported that NaHCO_3_ was reduced into formate with high efficiency by using Fe as a reductant [14]. The overall equation can be shown as:$$3Fe+NaHCO_{3}+3H_{2}O\to Fe_{3}O_{4}+3H_{2}(g)+\mathrm{HCOONa}$$
$$\Delta G\;\left(573K\right)=-132.776\mathrm{kJ}\cdot {\mathrm{mol}}^{-1}$$

A theoretical study was conducted by using the DFT calculations [49]. Under hydrothermal reaction conditions, the in situ technique for experimentally studying the reaction mechanism of formate production is very challenging.

#### 5.3.1. Optimized Geometric Parameters

The geometric parameters of the transition state and the final state are shown in Figure 16. The Fe–H bond distance in the transition state was approximately 1.505 Å, and the Mulliken charge of H in the iron hydride species was 0.029, as shown in Figure 16. The distance between C and H atoms was 1.569 Å. Compared to the distance in the intermediate, the distance is closer. The distance between metal and H atoms is similar to our previous study on CO_2_ reduction by zinc, which was 1.490 Å. The Fe–O bond distance was approximately 1.85~1.86 Å. In the final state, the bond distance of C-H was 1.107 Å. The Mulliken charge of H in the C–H species was 0.099. Compared with the distance in the intermediate and the transition state, the distance of C-H bond was closer, at only 1.107 Å. It means that the C-H bond is stronger, and the final state is more stable. The Fe–O bond distances were approximately 1.8~1.9 Å, and the charge of Fe was 0.674. The bond angle between the H-C-O bonds was 115.5°. Based on these bond distance results, we proposed the mechanism of HCOO production, as follows. In the hydrogen production process, the Fe–H^δ−^ species was formed via the dissociation of water. Due to the high reducibility of H^δ−^, the distance between H and C atoms became closer. When a transition state during the attack of H^δ−^ on C^δ+^ forms, the HCOO species can be subsequently produced.

#### 5.3.2. Energy Diagram for HCOO Production

An energy diagram of the C-H bond formation is shown in Figure 17. The optimized geometry of the initial, intermediate, and final states is depicted in Figure 16. As shown in Figure 17, the calculated activation energy is 16.43 kcal/mol. Compared to our previous study, when HCO_3_^−^ was used as the initial state, the calculated activation energy was 24.1 kcal/mol, which means that the CO_2_ was easier to reduce than HCO_3_^−^ under reaction conditions. An important implication of the calculated activation energy is that the in situ produced Fe-H species has high activity regarding reduction properties.

#### 5.3.3. IRC and HOMO/LUMO Calculation of Transition State

The IRC calculations and the HOMO and LUMO orbital shapes of the transition state (TS) are shown in Figure 18.

The IRC curve is smooth, as shown in Figure 18a. The formate was produced in the vertex of the curve. It can easily be seen that the reaction energy barrier was not very high, which means that formate could be easily produced once the Fe–H species was formed. As shown in Figure 18b,c, the occupied HOMO of the transition state exhibits a bonding interaction between the C atom and the H atom of the iron hydride species. On the other hand, the unoccupied LUMO of the transition state shows an anti-bonding character. These results clearly show the formation of a C-H bond of the formate. From the mechanistic points of view, the present findings provide new chemical insight to understand the hydrothermal reduction of CO_2_ to formic acid by metal. Additionally, these findings would benefit the understanding of similar reactions.

## 6. Hydrothermal Reduction of CO_2_ to Methanol

At present, the shortage of petroleum resources and the emission of CO_2_ coming from the consumption of petroleum resources are paid much attention. Therefore, it is urgent to develop C1 chemistry for substituting the feedstock of petroleum chemicals and supplying clean fuel. Methanol is an important C1 chemical. Methanol and its derived products can be used in gasoline, diesel oil, and domestic fuel. The investigation on the supply and demand of the methanol market proves that methanol is not only a chemical feedstock but also a clean energy. Therefore, the conversion of CO_2_ to methanol is very meaningful. Over the past several decades, CO_2_ conversion to methanol has been studied extensively because of its abundance and low cost. As we know, CO_2_ has high thermodynamic stability, therefore the catalysts are necessary for highly efficient CO_2_ reduction. Cu-based catalysts have always been utilized in the methanol production process [50,51,52]. Therefore, the hydrothermal reduction of CO_2_ to methanol has always been related to the development and utilization of Cu-based catalysts.

For example, Huo et al. reported the hydrothermal reduction of CO_2_ to methanol over copper as a catalyst [53]. As shown in Figure 19, the methanol yield (11.4%) was achieved at 350 °C for 3 h. When the amount of catalyst Cu was added, there was a small increase of the methanol yield. With a higher ratio of Zn and NaHCO_3_, the amount of reductant is surplus, and the methanol production increased. However, compared to the reaction temperature of formate production, methanol production requires a higher temperature, which means that the formation of methanol needs higher active energy. Therefore, a suitable catalyst should be developed to decrease the reaction temperature and increase the production efficiency.

The reaction mechanism was proposed subsequently. As shown in Figure 20, initially, CO_2_ is adsorbed on ZnO. Metal hydrides were produced based on the dissociation of hydrogen, which induced the methoxide species. As we know, Cu is a familiar catalyst for the production of methanol. Here, we suggest that the methanol was produced based on the formation of formate. When HCOO species was formed, the metal hydrides further accelerated the hydrogenation of HCOO, which produced an additional two C-H bonds. However, the detailed mechanism of methanol needs more information of intermediates and more detection methods, which should be discovered by in situ conditions.

Kattel et al. investigated the production methanol by Cu and ZnO via the formate intermediates [54]. It was concluded that there are three key steps for the formation of methanol: (1) the change of morphology with Cu under different conditions; (2) the formation of ZnCu alloy; and (3) ZnO_X_ overlayer on the Cu nano particles. Hu et al. studied the CO_2_ reduction to methanol under hydrothermal conditions [55]. By using the in situ Fourier transform infrared spectroscopy (FTIR) method, ex situ X-ray photoelectron spectroscopy, and high sensitivity low energy ion scattering spectroscopy, the intermediates were identified, as shown in Figure 21. In situ FTIR spectra indicated that the 5 wt.% Cu/A1_2_O_3_ can absorb large amounts of bicarbonate and carbonate species. Cu nano particles can effectively dissociate H_2_. Then, adsorbed bicarbonate or carbonate species can convert into formate and subsequently convert into a methoxy species. The peak at 1710 cm^−1^ can be the Zn-H bond, which further demonstrated when Methoxy species can finally be formed. When the methanol was produced, it was found that the peaks of formate and carbonate were decreased (see Figure 21). The founding gives us some useful information for methanol formation based on metal hydrides.

## 7. Hydrothermal Reduction of CO_2_ to Methane

Methane is an important starting material for the synthesis of numerous chemicals. The most important and common products include ammonia, methanol, acetylene, and synthesis gas. Methane is used in the petrochemical industry to produce synthesis gas, which is then used as a feedstock in other reactions. In recent years, CO_2_ reduction to methane has attracted increasing interest. Although it has been proved to be promising, industrial utilization is still a big problem due to the low yield, strict reaction condition, and high cost. Fischer−Tropsch synthesis is commonly used in CO_2_ conversion to value-added hydrocarbons [56,57]. It was found that iron could be used for CO_2_ reduction to methane and other products [58,59]. Developing a new method to effectively convert CO_2_ into methane was strongly desired.

Recently, Feng et al. performed a serial study on the hydrothermal reductions of CO_2_ to CH_4_ by using iron nano particles [60]. The proposed process of methane formation is as follows:$$CO_{2}+H_{2}O\overset{\mathrm{iron}(\mathrm{cat}.)}{\to }CH_{4}$$

As shown in Figure 22, the yield of CH_4_ increased with an increase of reaction time. The yield reached a maximum value of 1.96 mol% and then stably. When the reaction temperature increased higher, the yield increased significantly. Apparently, CO_2_ was more activated under higher temperature, which is favorable for the reduction of CO_2_. The higher pressure was also beneficial for the reduction by the increase of adsorption of CO_2_ on the surface of the iron nano particles. Compared to our previous study on the formic acid production, the yield of methane was low. The reason may be that the reaction temperature was lower than that for the production of formic acid. In future work, experiments with high reaction temperature should be performed to verify the possibility of the increase of methane production.

Based on the experimental results, the possible reaction mechanism for the formation of methane was proposed. As shown in Figure 23, iron reacted with water, generating H_2_ gas, and simultaneously, CO_2_ was reduced on the surface of iron nano particles due to the hydrogenation effect of hydrogen. Without the addition of iron nano particle, no methane was produced, which means that the addition of iron is necessary. As mentioned above, when iron reacted with H_2_O, hydrogen gas was produced, therefore, the formation of methane was related to the metal hydrides. Although, in this article, there was almost no information about the intermediates, we suggested that the hydrogenation was carried out by the intermediate metal hydrides. As far as we know, there is still a lack of theoretical study on this process to date. Because of the good prospects of methane utilization in the energy consumption area, the hydrothermal reduction of CO_2_ to methane should be studied further.

Zhong et al. reported a new method of in situ synthesis of Ni nanoparticle catalyst for the conversion of CO_2_ into methane [61]. An excellent 98% yield of methane was obtained at 300 °C. The in situ-formed Ni nano particle catalyst exhibits high efficiency for the production of methane from CO_2_. Additionally, the stability of the catalyst is still satisfied. The metal reductants (such as, Fe, Zn, etc) were oxidized into oxides that could be regenerated by using renewable energy or material [62]. Given these merits, this process developed a new concept for highly efficient artificial photosynthesis.

The methane yield from gaseous CO_2_ at different reaction conditions is shown in Figure 24a,b. As shown in Figure 24, the yield of methane increased while increasing the reaction time, temperature, and reductant amount. The highest yield of methane was only 42%, which is much lower than that with Zn. The yields of methane with CO_2_ or HCO_3_^−^ were investigated (see Figure 24c). The yield of methane obtained from HCO_3_^−^ was lower than that from the gaseous CO_2_. An excellent 98% yield of methane from the gaseous CO_2_ was obtained with the initial pH of 1. The catalytic activity of the Ni nanoparticle remained stable under high temperatures. These results suggested that the highly efficient reduction of CO_2_ to methane is promising with the addition of Ni as the catalyst.

## 8. Hydrothermal Reduction of CO_2_ to Long-Chain Hydrocarbons

CO_2_ utilization has received considerable attention due to its promising effect in reducing the carbon footprint and achieving carbon balance [63,64,65]. In general, the abiotic formation of hydrocarbons is assumed through the Fischer–Tropsch synthesis pathway [66,67]. The CO_2(aq)_ or HCO_3_^−^ (>70%) react with dissolved hydrogen H_2(aq)_ to produce hydrocarbons. Cobalt–iron alloys have been found in the Earth’s serpentinized ultramafic rocks, which verified the possibility for hydrocarbons production [68,69]. Thus, the abundant native iron and other transition metals are considered as promising reductants and/or catalysts for the natural production of hydrocarbons. Methane is the first alkane, which carries the suffix “ane”, denoting an alkane. Carbon is at the center of the tetrahedron, which can be assumed to be an equilateral pyramid. Methane is the principal component of natural gas, which contained at least 75% methane. It should be noted that the shortage of energy and the emissions of CO_2_ are big problems for humankind. Therefore, the reduction from methane to higher hydrocarbon is very important. Recently, Jin et al. reported the hydrothermal synthesis of long-chain hydrocarbons with Fe as a reductant and Co as a catalyst [70]. Interestingly, it was found that CoO_x_ can be reduced via CoCO_3_ with bicarbonate under hydrothermal conditions, coupled with the production of long-chain hydrocarbon under a temperature of about 300 °C and a pressure of about 30 MPa. High catalytic activity of Co for C–C coupling was verified by the formation of long-chain hydrocarbons. As shown in Figure 25, the products were mainly a series of homogenous linear alkanes with carbon chains of up to 24 carbons. Additionally, branched alkanes were also detected. These results were very interesting, because it was assumed that the hydrocarbons were produced from biomass, including lignin and cellulose. Therefore, future work should be done to improve the yields of hydrocarbons and decrease the reaction conditions, such as the decrease of reaction temperature and pressure.

By using ATR-FTIR, two significant peaks at 3676 and 3226 cm^−1^ were detected, which were attributed to the surface hydroxyl groups on Fe (Fe-OH) (see Figure 26a). When H_2_ gas was used as the reductant instead of Fe, there was no long-chain hydrocarbon produced. Therefore, the metal is necessary for the formation of hydrocarbons. In addition, FeCO_3_ was observed, as shown in Figure 26b. Figure 26c presents the distribution of the products as the function of the additional Co amount. It was found that the formation of HCOO^−^ gradually decreased with the increase of Co amount. Therefore, it can be concluded that the presence of Co reduced the formation of HCOO^−^ and promoted the formation of hydrocarbons. Thus, it was suggested that the formate formation is related to the production of hydrocarbons.

Based on the mechanism of C−C coupling, a possible pathway of NaHCO_3_ reduction to hydrocarbons with Fe and Co was proposed, as shown in Figure 27. The proposed mechanism is similar to the abiotic synthesis in an actual hydrothermal environment [71]. The reaction system has both similar hydrothermal conditions and reaction pathways, which suggests the abiogenic origin of some hydrothermal petroleum. However, in our opinion, the reaction pathways may be complicated in nature, which means that the formation of the final products may come from a different reaction pathway. The formate production may be the key step. As we introduced above, the yield of formate was very high under metal–water hydrothermal conditions. The hydrogenation step, subsequently, was also very important, as it induced the C–H bond formation one by one. Eventually, the methane was produced. For the C–C coupling, the detailed mechanism still was not clear, and requires further study, including experimental and theoretical study.

## 9. Conclusions

Hydrogen production from H_2_O dissociation under hydrothermal conditions is one of the most ideal methods due to its environmentally friendly impact. The hydrogen could be produced rapidly under hydrothermal conditions with metals. Some recent experimental studies on the hydrothermal reduction of CO_2_ by in situ generated metal hydrides, including Zn, Al, and Fe, were introduced. IR studies verified the existence of metal hydrides, which suggested that the metal hydrides provided active hydrogen for the hydrogenation of CO_2_. In the theoretical studies, the DFT calculations indicate that the metal hydrides are important intermediates that can reduce CO_2_ to formate through an SN2-like mechanism. With the catalysts Cu and Co, the methanol, methane and long-chain hydrocarbons could be produced. As a new reaction system, the primary experimental results have been acquired. However, the detailed reaction mechanism still needs further study. Then, an integrated process by coupling hydrothermal reactions, including CO_2_ reduction and H_2_O dissociation with metals, is proposed. These studies indicate that the hydrothermal reduction of CO_2_ by in situ generated metal hydrides is very promising, not only for the hydrogen storage to formate, but also for the CO_2_ conversion to useful chemicals and fuels, which would help facilitate solutions for global warming and the energy crisis.

## 10. Outlook

In the 21st century, global warming and energy shortages have become worldwide problems and challenges. For example, in recent several years, flood disasters appeared around the world, which, induced by climate change, have influenced many people’s lives. The problem of CO_2_, which comes from the consumption of fossil fuels, induced the greenhouse effect and energy shortages. The urgent need to reduce the CO_2_ concentration in the air has prompted action all over the world.

Future work to accelerate the development of hydrothermal reduction of CO_2_ includes:(1)developing a stable catalyst to increase the reaction efficiency and decrease the reaction temperature;(2)exploring the detailed reaction mechanism to enhance the understanding of the reaction process;(3)developing reaction devices to perform industrial processes.

## Data Availability

The data presented in this study are available on request from the corresponding author.

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
