# Peer review of "Hydrothermal Reduction of CO2 to Value-Added Products by In Situ Generated Metal Hydrides"

_materials, 2023, doi:10.3390/ma16072902_

Round 1
Reviewer 1 Report
The topic of the manuscript entitled: Hydrothermal Reduction of CO2 to Value-added Products by in situ Generated Metal Hydrids is very interesting and much needed in the recent time more than ever.
The authors may be too much involved in the processes reviewed in this manuscript. However, it is impossible for a reader to follow the text as the authors do jump from a process to another without a clear structure and without giving suffiecient information on each cited process.
Strictly speaking, the following points are major shortcomings, which need extensive editing revision before a new submission:
1) There should be a complete English revision for the manuscript, which is full of grammatic and typing errors.
2) A generic process layout should be developed to cover all processes described throughout the manuscript, which can be reduced to the relevant processes by neglecting the items, which are not involved in the cited reference(s). The generic process can e.g. replace Scheme 1, which says alomst nothing for the reader.
3) Specific information on the source of CO2 (captured out of ambient air, or from a CO2 bottle, or even mixed with other gases) must be included for each cited reference. The methodology and the sequence of each process shall be described in more details, along with the integrated sensors and actuators.
4) to give an example, the description of Figure 3 in lines 112-122, the given numbers are not matching with the related curves inside the Figure. It should be also distinguished between the left-side (a) and the right-side (b) diagrams, means a sub-title must be included for each and in the text, both Figures should be referred to.
5) Another example for the style, which is not allowed in a scientific paper (of course in a newspaper essay it is allowed) is what is written in sentences 131-133:
"These results illustrated that zinc could be reaction with water efficiently, which means that the efficiency was very high. Therefore, from the industrial utilization, the energy cost would be very favorable."
The red highlithed text is what I mean; namely, very high and very favorable are too general description for a scientific work. Instead specific numbers much be given to justify such descriptions.
6) Another example is the text in lines (493-494) which must be reviewed completely. In this paragraph, the process conditions (pressure and temperature) are not methioned specifically but both were described with "high" or "higher", a style which must be avoided in the revised mansucript and specific values should be given instead.
Reviewer 2 Report
File attached.

Reviewer 3 Report
The manuscript was written based on author's inspiration on coupling hydrothermal H2O dissociation and CO2 reduction to synthesize fuels and chemicals (i.e., acid formic, methanol, methane, long chain hydrocarbons) in the presence of metals.
1) The quality of English throughout the manuscript requires extensive improvement.
2) It is unclear to me on whether the idea to couple the two reactions was from the author, or from other authors. It appears to me the idea was coming from elsewhere (ref [14]), which is not author's original idea.
3) The presentation of the two reactions to produce fuels/chemicals are not well connected (the presence of metal oxides and metal hydrides in the mechanism seems disconnected) and I'm unable to appreciate the idea of coupling them for the application.
4) Each section must be supported with author's comment on how the findings from a certain work can be useful in the proposed idea. Else, they only serve as info.
Once these major concerns are addressed, then it would be easier for me to review further. At its current form, it is not a convincing piece of work for publication.
Round 2
Reviewer 1 Report
All reviewers' comments are adressed quite well in th erevised version. The remaining minor english corrections are left to the editorial team.
I recommend the puplication of the revised manuscript after the english approval of the editorail team.
Wish the authors a sound success
Author Response
Thank you very much for your kind review letter.
In accordance with your comments, we checked the manuscript again and revised carefully.
Thanks again!
Reviewer 3 Report
I truly appreciate the efforts put in revising the manuscript. I can totally understand that the manuscript is written by a student who is lack of experience in writing scientific papers. However, I expect that the more experienced co-authors in the team would have played their role to provide the right feedback and guidance in the manuscript preparation and correction process, but this is not reflected in the manuscript.
1) The manuscript still suffers from very low quality of English and the writing style is not suitable for publication in a reputable scientific journal.
2) The title suggests that metal hydrides are responsible as the active material in hydrothermal reduction of CO2, but this is not reflected in the manuscript for all the cases.
3) Scheme 1 shows that the CO2 is derived from human consumption. To me, the focus should be more specific to industry emissions. Also, it is not shown where the H2O is involved in this scheme. If I understand it correctly, the metal oxides as active material is being reduced to pure metal in H2O dissociation to produce hydrogen, after which metal hydrides are produced for CO2 reduction. So why is the arrow showing that that pure metal is reoxidized to metal oxides. This is confusing. Also, in this Scheme, where is metal hydrides, which should be the main focus as the active material for hydrothermal reduction of CO2 as suggested by the title?
4) Section 2.2, how is water density an important property for the reaction? Is it related to section 2.4? If it is, then connect them.
5) No reference was cited in section 4. This is only an example of may other parts.
6) A lot of what being written in the manuscript are irrelevant, unstructured, and disorganized making the manuscript very difficult to follow.
7) Line 161, what is species H-Zn…O-H??
8) Line 168 – 170, “Therefore, the IR absorption peak indicates that the zinc hydrides were produced as the intermediates, which gives some information for the reaction mechanism explanation.” Which reaction mechanism? I didn’t see any mechanism steps shown in the manuscript.
9) The reporting doesn’t fit the style of a review paper. It appears as the author taking the work of others and still report it in a style of an original article, trying to re-produce the same conclusions as the original papers. Example:
“Although the formic acid yield increased with longer time, 2 h is a better choice for considering the energy cost. As shown in Figure 7, the results also showed that formic acid yields increased significantly when the reaction temperature increased from 250 to 300 ℃. It was very interesting that the yield decreased to 34% at 325 ℃. The reason is probably that the decomposition of formic acid speeded up under higher temperature. Therefore, the preferred reaction temperature is 300 ℃. The variation trend of the formic acid yields with Al is similar with the results of Zn. The reason for that may be the decomposition of the formate under higher temperature with Al and Zn. Therefore, it is better to control the reaction temperature at 300 ℃.”
10) Absence of scientific soundness
“For the reaction time, maybe the reaction time 2 h is a better choice. In our previous study on the hydrothermal oxidation of biomass, we also found that the suitable reaction temperature was also at 300 ℃. Probably it was related to the special characteristics of water under high temperature.”
The choice of word “maybe” is already suggesting poor scientific soundness of the sentence. This is only an example of many others. Also, why would it be necessary for the author to conclude the original reference on the better choice of the reaction time? What’s the significance? This is an example of weird writing style of a review article.
